# Node properties of biomarkers within the protein–protein interaction network derived from breast cancer-associated genes

**Takanori Sasaki** **\*, Saito Torii**

Department of Network design, Graduate School of Advanced Mathematical Sciences, Meiji University, Tokyo, Japan

\* tsasaki@meiji.ac.jp

## Abstract

Analyzing the network properties of cancer biomarkers within protein–protein interaction (PPI) networks is valuable for discovering novel biomarker candidates. Therefore, we constructed PPI networks using breast cancer (BC)-associated gene sets and performed 12 distinct centrality analyses to characterize the topological features of clinically validated biomarkers. Our reference set of biomarkers comprised genes from five clinical genetic testing panels—MammaPrint, Oncotype DX, PAM50, EndoPredict, and the BC Index—that were also present in the STRING database. The PPI networks were constructed from the top 2,000 BC-associated genes, ranked by disease score from the DISEASES database. These networks were then subjected to centrality analysis using five local and seven global measures. The top 5% centrality rankings were evaluated, demonstrating that maximum clique centrality (MCC) identified the highest proportion of known biomarkers, with an inclusion rate of approximately 36%. Furthermore, MCC generated a unique biomarker-ranking pattern, exhibiting a Spearman's rank correlation coefficient below 0.8 when compared with all other metrics. Consequently, a high MCC score is a key topological feature of many validated biomarkers. Genes with the highest MCC scores (top 5%) were significantly enriched for gene-ontology terms related to the cell cycle and fibroblast growth factor receptor signaling pathway. Additionally, biomarkers with high MCC scores exhibited significantly greater evolutionary conservation and potential for protein complex formation. Collectively, our findings indicate that many effective BC biomarkers are components of large, evolutionarily conserved cliques within cell-cycle-associated regions of the PPI network. Finally, based on this MCC-centric approach, we identified 11 novel candidate biomarkers.

**Data availability statement:** All data used in this study are publicly available. Breast cancer–associated genes were retrieved from the DISEASES database (https://diseases. jensenlab.org) using the disease query terms "breast cancer" (DOID:1612), "breast disease" (DOID:3463), and "breast carcinoma" (DOID:3459) (accessed February 23, 2024), yielding 5,646 genes ranked by disease score. The top 2,000 genes were selected based on disease score (disease score >2.08) using the Cytoscape plugin stringApp (v2.0.3), and 1,816 genes present in the STRING database (https://string-db.org) were used for protein–protein interaction network construction. PPI networks were generated using STRING with interaction sources "Experiments," "Databases," and "Co-expression," and minimum interaction scores of 0.4, 0.7, and 0.9. Functional enrichment analysis was performed using Metascape (https://metascape.org) on the top 5% of MCC-ranked genes (n = 68). Survival analysis was conducted using the Kaplan–Meier Plotter (https://kmplot.com), and gene expression comparisons were performed using TNMplot (https://tnmplot.com) on 45 candidate genes. Ortholog information was obtained from InParanoidDB v9 (http://inparanoid.sbc.su.se) using the "full-length orthologs" search option.

**Funding:** The author(s) received no specific funding for this work.

**Competing interests:** The authors have declared that no competing interests exist.

**Abbreviations:** BC, Breast cancer; CC, Clustering coefficient; DMNC, Density of maximum neighborhood component; EPC, Edge percolated component; MNC, Maximum neighborhood component; MCC, Maximum clique centrality; PPI, Protein-protein interaction; GO, Gene ontology; RFS, relapse-free survival; OS, overall survival.

## Introduction

Breast cancer (BC) is the most frequently diagnosed malignancy in women and a leading cause of cancer-related mortality [1]. The clinical management of BC is challenged by its heterogeneity, making an understanding of the specific mutations and gene expression changes within individual tumors essential for developing effective therapeutic strategies [2,3]. In this context, biomarkers function as critical tools for diagnosis and prognostication. For example, established multigene tests such as Oncotype DX, PAM50, MammaPrint, EndoPredict, and the BC Index have been developed and validated by correlating gene signatures with patient clinical outcomes [4–8]. Each of these assays has demonstrated robust prognostic power, contributing significantly to personalized treatment and improving patient outcomes [9].

Meanwhile, numerous computational biology studies have focused on identifying novel biomarker candidates for BC [10,11]. These approaches have often employed gene co-expression analysis and machine-learning techniques to identify candidate genes and subsequently assess their prognostic value on large datasets and through in vitro experiments. However, no prior studies have explored novel biomarker candidates by leveraging the computational characteristics of biomarkers already established in clinical practice.

Protein–protein interaction (PPI) networks are invaluable for elucidating both the maintenance of cellular functions and the roles of genes in disease progression. Centrality analysis, a tool from network theory, has been widely applied to characterize disease-associated proteins as nodes within these PPI networks [12–14]. For instance, Viacava Follis (2021) reported that targets of available selective drugs tend to possess a high degree of centrality, short average path lengths, and low topological coefficients in the PPI network [13]. In networks of prostate cancer-associated genes, "kinless hubs" that bridge modules were identified as essential for maintaining network stability and integrity [15]. Similarly, a centrality analysis of BC-related PPI networks showed that these genes often exhibit high clustering coefficients and link densities [16]. This body of work confirms that disease-associated genes possess distinct node characteristics. However, it remains unclear whether clinically implemented biomarkers for BC share common centrality features within disease-specific PPI networks.

This study was designed to investigate the node properties of established BC biomarkers in a PPI network context. We constructed networks using BC-related genes with high disease scores from the DISEASES database and applied 12 centrality analyses. Our investigation revealed that high maximum clique centrality (MCC) scores are a characteristic feature of many BC biomarkers. Based on this finding, we used the optimal network conditions for MCC to propose 11 new candidate biomarkers, providing an analytical framework to advance network-based biomarker discovery.

## Materials and Methods

### Data acquisition and preprocessing

The workflow of this study is illustrated in Fig 1.

Genes associated with BC were obtained from the DISEASES database, which links genes to diseases via a confidence metric called a disease score (scale: 0–5) [17,18]. Using the Cytoscape plugin stringApp (v2.0.3), we queried the database for "BC," which returned an initial set of 5,646 genes [19]. The database was accessed on February 23, 2024. Because DISEASES is continuously updated, the analyses in this study reflect the database version available at the time of access. From this list, we selected the top 2,000 BC-related genes, resulting in a list with disease scores above 2.08.

## PPI network construction using BC-related genes

A PPI network was constructed using the selected BC-related genes that were registered in the DISEASES and STRING databases [20]. To define the network edges, physical and functional interactions were sourced from STRING by selecting "Experiments," "Databases," and "Co-expression" as interaction sources. The density of each resulting PPI network was calculated using the formula $2m/n(n-1)$, where $m$ and $n$ denote the numbers of edges and nodes, respectively.

## Centrality analysis

Centrality analysis was performed on the BC-related PPI network with the cytoHubba (v0.1) plugin for Cytoscape [12]. We calculated 12 centrality measures: Betweenness, BottleNeck, Clustering Coefficient (CC), Closeness, Degree, Density of Maximum Neighborhood Component (DMNC), EcCentricity, Edge Percolated Component (EPC), MCC, Maximum Neighborhood Component (MNC), Radiality, and Stress. The biomarker set for this analysis comprised genes from five genetic tests (MammaPrint, Oncotype DX, PAM50, EndoPredict, and the BC Index) that were also registered in the STRING

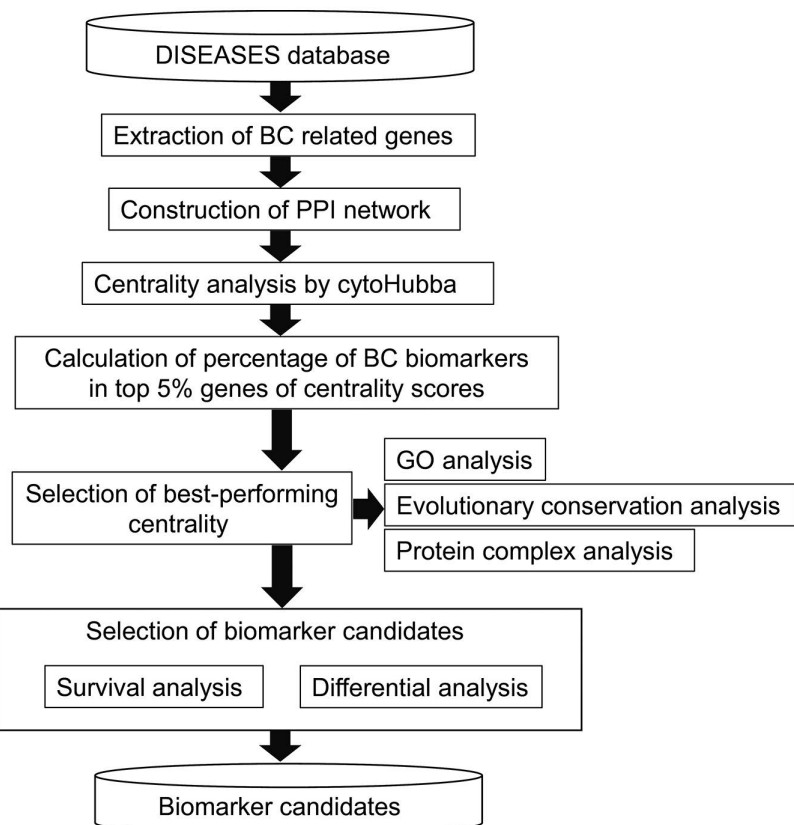

**Fig 1. Study workflow.** BC: Breast Cancer, PPI: protein–protein interaction.

database. Using the resulting centrality rankings for these biomarkers, we calculated Spearman's rank correlation coefficients using the R package corrplot (v0.92) and performed hierarchical clustering via Ward's method.

### Gene ontology (GO) enrichment analysis

GO enrichment analysis was conducted on genes with high-ranking centrality score using Metascape (v3.5) [21]. A custom analysis was conducted using GO biological process-related genes from *H. sapiens* as the background set. To explore the semantic relationships between the resulting GO terms, we used REVIGO (v1.8.1) [22,23]. These results were then exported as an R script for plotting.

### Survival analysis

Survival analyses of BC-related genes were performed using the Kaplan–Meier Plotter public datasets [24]. Patients were stratified into two groups based on the median expression level of a given gene, and analyses were conducted for both relapse-free survival (RFS) and overall survival (OS). A $p$-value $< 0.05$ was considered statistically significant.

### Differential expression analysis

Differential gene expression was analyzed using the TNMplot tool, which contains data from 56,938 unique samples [25,26]. To compare gene expression across normal, tumor, and metastatic tissues, the Kruskal–Wallis test was applied, followed by Dunn's test for post hoc comparisons. A $p$-value $< 0.05$ in the Dunn's tests for normal vs. tumor and tumor vs. metastatic samples was considered statistically significant.

### Ortholog pair analysis

To evaluate the evolutionary conservation of BC biomarkers, we identified orthologous pairs using InParanoidDB9 [27,28]. Pairwise comparisons were performed between *H. sapiens* and five other eukaryotic species: *S. cerevisiae, D. melanogaster, C. elegans, A. thaliana, and D. rerio*. Seed ortholog pairs were defined as genes from full-length protein ortholog groups with an Inparalog score of 1.0 and a seed score $\geq 0.95$ from the InParanoid-DIAMOND algorithm. Based on this, an evolutionary conservation score was assigned to each biomarker, with 0.2 points awarded for the presence of an ortholog in each species, for a maximum possible score of 1.0.

### Prediction of complex formation ability

The potential for BC biomarkers to form protein complexes was predicted using the MCODE (v2.0.3) plugin for Cytoscape [29]. The analysis was run with default parameters on the BC-related PPI network described in Section 2.2. The MCODE-generated node score, which is used to identify cluster "seed" proteins, was used as a measure of each protein's ability to form complexes.

## Results

### PPI network construction using BC-related genes

To generate a series of PPI networks, we first obtained the top 2,000 BC-related genes ranked by disease score from the DISEASES database. From this list, we created three sets of "query genes" using the top 1,000, 1,500, and 2,000 genes. These sets corresponded to 920, 1,362, and 1,816 genes, respectively, that were registered in the STRING database (Table 1).

For each gene set, networks were constructed using three different interaction score cutoffs: $> 0.4$ (medium confidence), $> 0.7$ (high confidence), and $>0.9$ (highest confidence). For example, the PPI network built from 1,000 query genes with medium confidence interactions consisted of 920 STRING nodes and 11,032 edges, as visualized in Fig 2(A).

**Table 1. Construction of PPI network based on BC related genes obtained from DISEASES database.**

| BC related genes from DISEASES database | | PPI network from STRING database[a] | | | |
|---|---|---|---|---|---|
| Number of query genes | Disease score | Nodes[b] (Biomarkers)[c] | Edges (Interaction score)[d] | | |
| 1,000 | >2.39 | 920 (46) | 11,032 (0.4) | 4,039 (0.7) | 2,608 (0.9) |
| 1,500 | >2.21 | 1,362 (64) | 17,907 (0.4) | 6,419 (0.7) | 3,973 (0.9) |
| 2,000 | >2.08 | 1,816 (72) | 26,856 (0.4) | 9,274 (0.7) | 5,761 (0.9) |

[a]Active Interaction sources: Experiments, Databases, Co-expression.

[b]BC related genes registered in the STRING database.

[c]Number of biomarkers forming the nodes of PPI network.

[d]Minimum required interaction scores: 0.4 (medium confidence), 0.7 (high confidence), 0.9 (highest confidence).

BC: breast cancer, PPI: protein-protein interaction.

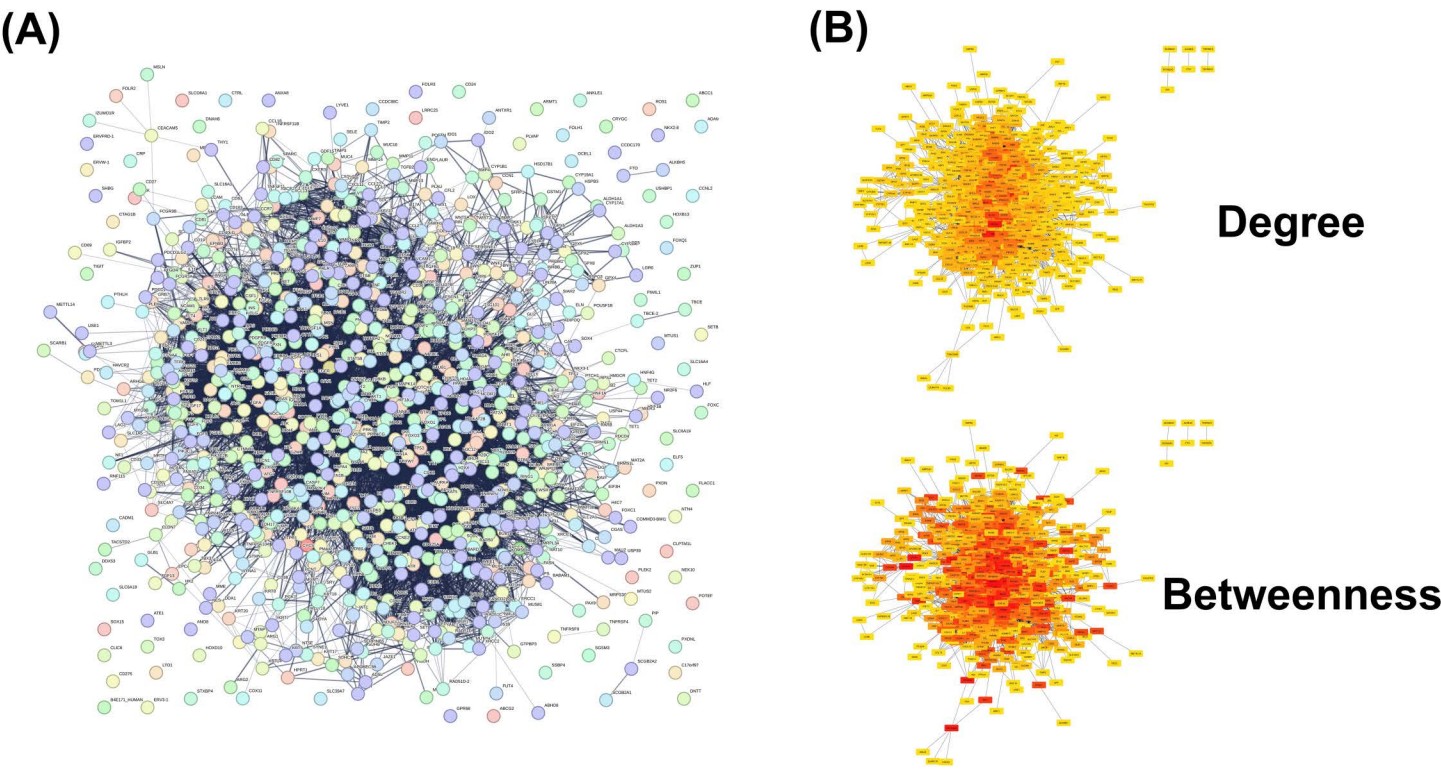

**Fig 2. PPI network of BC-related genes and centrality analysis. (A)** PPI network constructed with 920 nodes from the STRING database, derived from 1,000 query genes. Medium confidence edges (minimum required interaction score = 0.4) were applied. Active interaction sources were experiments, databases, and co-expression. **(B)** Results of centrality analysis. The upper panel shows scores for the local-based "Degree" method; the lower panel shows scores for the global-based "Betweenness" method. Deeper red node colors indicate higher centrality scores. Nodes with a degree of zero are not displayed. BC: Breast Cancer, PPI: protein–protein interaction.

Our target list for analysis comprised 128 known biomarkers from five representative genetic tests (Mammaprint, Oncotype DX, PAM50, EndoPredict, and BC Index), which were also registered in STRING (S1 Table). Within the constructed networks, 46, 64, and 72 of these biomarkers were present among the 920, 1,362, and 1,816 STRING nodes, respectively (Table 1).

## Centrality analysis on the PPI network

We applied five local-based and seven global-based centrality ranking methods to the constructed PPI networks, calculating scores for all genes with a degree of one or more (S2 Table). The results for two representative methods, the local-based "degree" and global-based "betweenness," are shown in Fig 2(B), whereas S1 Fig displays the results for the other ten analyses.

For each metric, we ranked all genes in the PPI network by their score and then calculated the percentage of known biomarkers that appeared in the top 5% of the ranking. This proportion is displayed for each centrality score in Fig 3. As shown in Fig 3(A), the inclusion rate of biomarkers was sensitive to the network confidence cutoff.

For example, using the betweenness centrality method with medium confidence interactions, the biomarker inclusion rates were $4/46 \times 100 = 8.7\%$, $5/64 \times 100 = 7.8\%$, and $6/72 \times 100 = 8.3\%$ for the 1,000-, 1,500-, and 2,000-gene networks, respectively, with an average of 8.3%. Generally, biomarker rankings tended to be higher under medium and high confidence settings than under the highest confidence setting, except for the "clustering coefficient" and "radiality" measures. Notably, the "EcCentricity" metric exhibited poor resolution, with many genes sharing the same score; in these instances, all genes with the same score as that of the top 5% threshold were included in the analysis and normalized accordingly. Fig 3(B) compares biomarker inclusion rates across PPI networks built with different numbers of query genes. Each bar represents the average inclusion rate calculated across the medium, high, and highest confidence networks. For locally based methods such as Degree, DMNC, MCC, and MNC, the inclusion rate was 1.5–2.1 times higher in networks constructed from 1,500 and 2,000 query genes compared with the 1,000-gene network. Across all comparisons shown in Figs 3(A) and (B), MCC consistently yielded the highest biomarker inclusion rate. These results strongly suggest that a high MCC score is a shared characteristic among established BC biomarkers.

## Relationship between biomarker MCC rankings and PPI network statistics

Next, we examined the influence of PPI network statistics on the inclusion rate of biomarkers among the top 5% of MCC-ranked genes. As shown in Fig 4, this rate varied depending on the number of query genes and the interaction confidence used to build the network (see S3 Table for full network statistics).

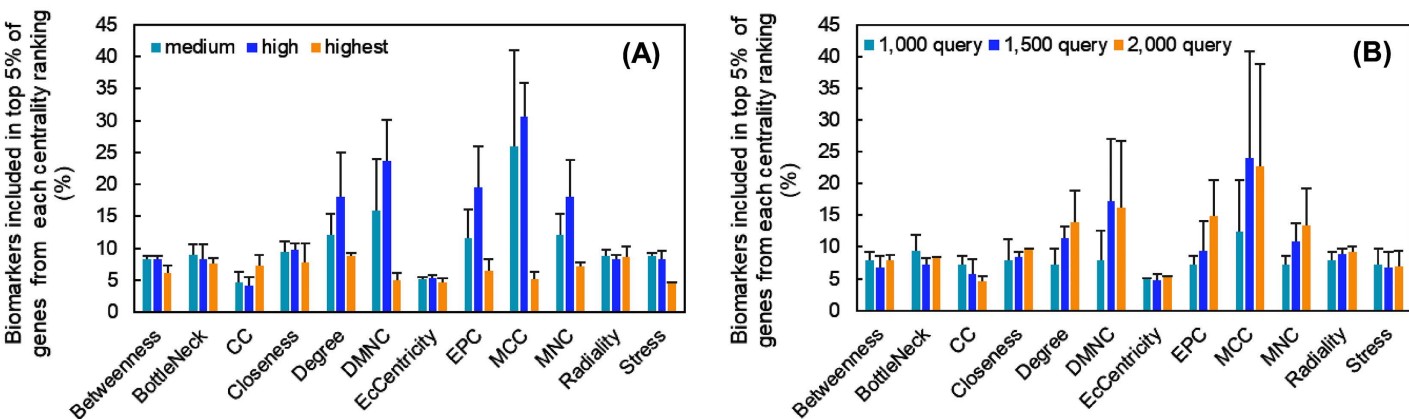

**Fig 3. Proportion of BC biomarkers included in the top 5% of genes from each centrality ranking.** (A) Centrality analysis results for PPI networks constructed with medium (light blue bar), high (blue bar), and highest (orange bar) confidence edges. Each bar represents the average proportion of biomarkers from networks built with 1,000, 1,500, and 2,000 query genes. Error bars indicate the standard deviation. (B) Centrality analysis results for PPI networks constructed with nodes from 1,000 (light blue bar), 1,500 (blue bar), and 2,000 (orange bar) query genes. Each bar represents the average proportion of biomarkers across all three confidence levels. Error bars indicate the standard deviation. BC: Breast Cancer, PPI: protein–protein interaction.

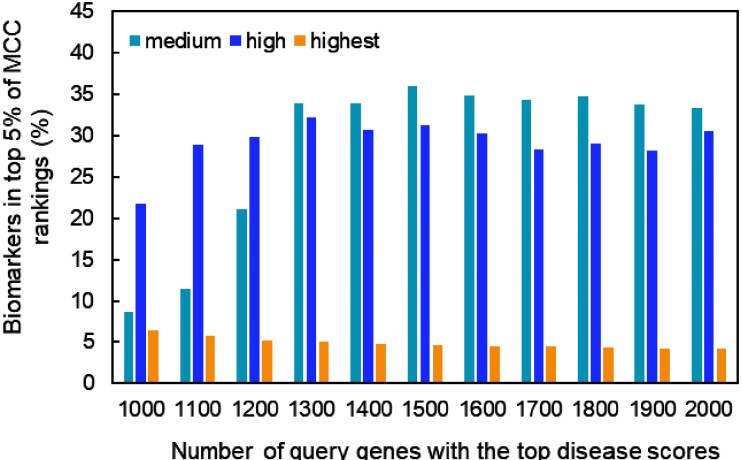

**Fig 4. Proportion of BC biomarkers in the top 5% of MCC rankings.** The results of MCC analysis are shown for PPI networks constructed from a range of 1,000 to 2,000 query genes with the top disease scores. Edge conditions were medium (light blue bar), high (blue bar), and highest (orange bar) confidence. BC: Breast Cancer, PPI: protein–protein interaction.

The optimal condition was the network constructed from 1,500 query genes (1,362 STRING nodes) with medium confidence interactions, which showed the highest inclusion rate, encompassing approximately 36% of the 64 biomarkers present in that network. To determine whether the MCC ranking pattern in this optimal network was distinct, we compared it with the other 11 centrality rankings for the 62 connected biomarkers (S4 Table; two unconnected genes, HOXB13 and SLC39A, were excluded. Fig 5 shows the Spearman's rank correlation coefficients and results of a hierarchical clustering analysis (correlation threshold = 0.8).

The analysis revealed three major clusters: one comprised global-based methods ("Betweenness," "Closeness," "Radiality," "Stress"), a second of local-based methods ("DMNC," "CC"), and a third that included "EPC," "Degree," and "MNC." By contrast, MCC, BottleNeck, and EcCentricity all had correlation coefficients below 0.8 with all other metrics, indicating that their ranking patterns are unique.

## GO of genes with high MCC scores

As the PPI network constructed from 1,500 query genes with medium confidence interactions showed the highest concentration of highly-ranked biomarkers, we performed a GO analysis on the top 5% of MCC-ranked genes from this network. This set comprised 68 genes, including 23 known biomarkers (Table 2).

An examination of the top 20 enriched GO terms revealed that one of the most significant was the "fibroblast growth factor receptor signaling pathway" ($P = 3.9 \times 10^{-45}$) (Table 3).

Additionally, five GO terms related to the "cell cycle" were significantly enriched: "mitotic cell cycle process" ($P = 3.1 \times 10^{-37}$), "regulation of cell cycle process" ($P = 7.6 \times 10^{-29}$), "cell cycle phase transition" ($P = 1.2 \times 10^{-22}$), "regulation of G2/M transition of mitotic cell cycle" ($P = 1.6 \times 10^{-12}$), and "meiotic cell cycle" ($P = 2.7 \times 10^{-10}$). To explore the relationships between these terms, we used REVIGO to cluster them based on semantic similarity (Fig 6).

This process identified two major cell-cycle-related clusters. The first, represented by the most significant term "mitotic cell cycle process," included related terms such as "cell cycle phase transition" and "meiotic cell cycle."The second group, represented by "regulation of cell cycle process," included terms such as "regulation of cell division" and "positive regulation of chromosome segregation". These results suggest that genes with high MCC scores are predominantly involved in the fibroblast growth factor receptor signaling pathway and cell-cycle-related processes.

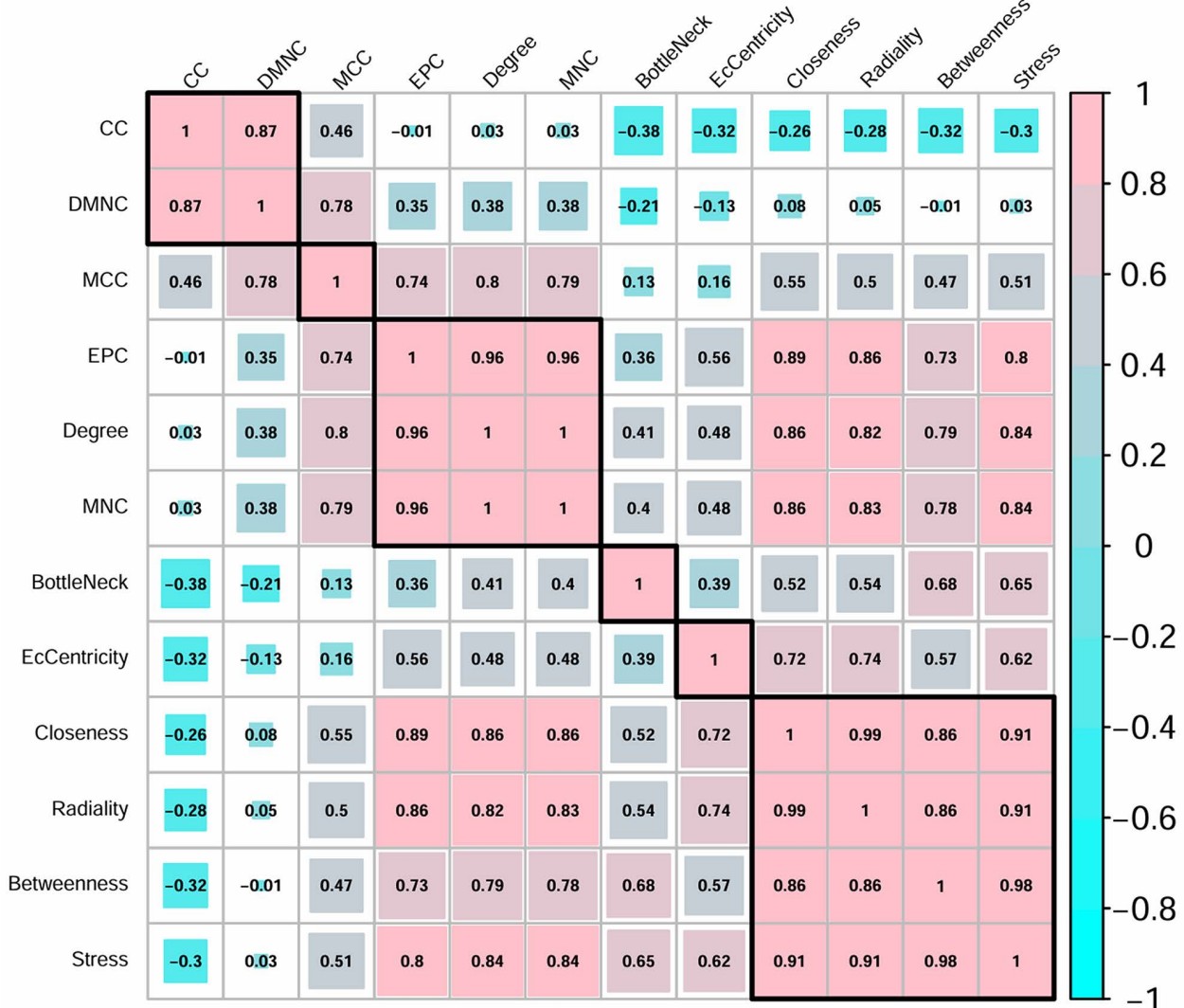

**Fig 5. Spearman's rank correlation coefficient and hierarchical clustering of centrality rankings for BC biomarkers.** The analysis used 62 biomarkers with a degree of ≥1 from the PPI network based on 1,500 query genes. Two biomarkers with a degree of 0 (HOXB13 and SLC39A) were excluded. BC: Breast Cancer, PPI: protein–protein interaction.

### Relationship between MCC rankings of biomarkers and their biological characteristics

To investigate the biological properties associated with MCC rankings, we first examined the evolutionary conservation of biomarkers. InParanoidDB9 was used to determine if *Homo sapiens* genes for each BC biomarker had orthologs in five representative eukaryotic species (*Saccharomyces cerevisiae*, *Drosophila melanogaster*, *Caenorhabditis elegans*, *Arabidopsis thaliana*, and *Danio rerio*) (S5 Table). A score of 0.2 was added to each biomarker's evolutionary conservation score for every orthologous pair that was considered a true ortholog, defined as having a seed score of ≥0.95. For instance, CDC6 (MCC rank 34), which has orthologs across all five species, received a maximum score of 1.0. A comparison of biomarkers divided into high- and low-MCC ranking groups (32 genes each) revealed that the high-MCC group had a significantly greater median conservation score ($P=0.02$, Wilcoxon rank-sum test; Fig 7(A), S7 Table).

**Table 2. Top 5% genes of MCC ranking for PPI network based on 1,500 query genes[a].**

| Rank | Gene name | Rank | Gene name | Rank | Gene name |
|------|-----------|------|-----------|------|-----------|
| 1 | CCNA2 | 24 | **MKI67** | 38 | FGF10 |
| 1 | **CCNB1[b]** | 25 | FOXM1 | 38 | FGF7 |
| 1 | CDK1 | 26 | **NEK2** | 38 | **FGF18** |
| 1 | **CDC20** | 27 | CDC25C | 38 | FGF19 |
| 5 | TTK | 28 | **UBE2C** | 38 | FGF16 |
| 5 | **AURKA** | 29 | PLK1 | 38 | FGF3 |
| 5 | TOP2A | 30 | **CENPF** | 38 | FGF2 |
| 8 | **MELK** | 31 | CDCA2 | 54 | PTPN11 |
| 8 | CCNB2 | 32 | **EXO1** | 55 | SHC1 |
| 8 | BUB1 | 33 | **PTTG1** | 56 | PIK3R1 |
| 11 | **BIRC5** | 34 | **CDC6** | 56 | PIK3CA |
| 12 | **KIF2C** | 35 | CHEK1 | 58 | **ERBB2** |
| 12 | TPX2 | 36 | TK1 | 59 | PIK3CB |
| 12 | **BUB1B** | 37 | GRB2 | 60 | KRAS |
| 15 | KIF11 | 38 | FGF4 | 61 | NRAS |
| 15 | **PRC1** | 38 | FGF20 | 61 | HRAS |
| 17 | **NDC80** | 38 | FGF6 | 63 | **FGFR4** |
| 18 | **CEP55** | 38 | FGF17 | 63 | FGFR3 |
| 19 | **RRM2** | 38 | FGF9 | 63 | FGFR2 |
| 19 | AURKB | 38 | FGF8 | 63 | FGFR1 |
| 21 | HMMR | 38 | FGF5 | 67 | **EGFR** |
| 22 | CDKN3 | 38 | FGF1 | 68 | ERBB3 |
| 23 | **CENPA** | 38 | FGF22 | | |

[a]STRING nodes of 1,362, medium confidence edges of 17,907.

[b]Bold refers to biomarkers.

PPI: protein-protein interaction.

Among the 11 other centrality metrics, only five showed a similar statistically significant difference, suggesting a strong association between high MCC rank and evolutionary conservation.

Furthermore, we investigated the link between biomarker MCC rankings and the potential for protein complex formation using the MCODE algorithm, a density-based clustering method for predicting protein complexes within PPI networks. MCODE calculates a node score for each protein, reflecting its local interaction density —a proxy for its likelihood of being part of a biological complex. Based on the scores for the 64 biomarkers (S6 Table), we found that those with higher MCC rankings had significantly higher node scores than biomarkers with lower rankings ($P = 4.0 \times 10^{-16}$, Wilcoxon rank-sum test; Fig 7(B), S7 Table). Although nine other centrality metrics also showed significant correlations, the association with MCC was the most statistically significant (S7 Table). These findings suggest that biomarkers with high MCC rankings, which are part of larger or more numerous network cliques, have a greater potential for protein complex formation.

### Prediction of novel biomarker candidates

Finally, we leveraged these findings to predict novel biomarker candidates by examining the top 5% of MCC-ranked genes, excluding those already known to be biomarkers. These potential candidates were filtered based on survival analysis in the Kaplan–Meier Plotter database (7,830 BC samples) and differential gene expression analysis in the TNMplot

**Table 3. GO term on top 5% of MCC-ranked genes.**

| No | Term ID | Term description | P value |
|---|---|---|---|
| 1 | GO:0008543 | fibroblast growth factor receptor signaling pathway | $3.9 \times 10^{-45}$ |
| 2 | GO:1903047 | mitotic cell cycle process | $3.1 \times 10^{-37}$ |
| 3 | GO:0010564 | regulation of cell cycle process | $7.6 \times 10^{-29}$ |
| 4 | GO:0044770 | cell cycle phase transition | $1.2 \times 10^{-22}$ |
| 5 | GO:0051338 | regulation of transferase activity | $7.6 \times 10^{-22}$ |
| 6 | GO:0051302 | regulation of cell division | $2.4 \times 10^{-21}$ |
| 7 | GO:0008283 | cell population proliferation | $2.0 \times 10^{-18}$ |
| 8 | GO:0016310 | phosphorylation | $3.9 \times 10^{-18}$ |
| 9 | GO:0030324 | lung development | $3.7 \times 10^{-13}$ |
| 10 | GO:0010389 | regulation of G2/M transition of mitotic cell cycle | $1.6 \times 10^{-12}$ |
| 11 | GO:0010001 | glial cell differentiation | $5.3 \times 10^{-11}$ |
| 12 | GO:0051321 | meiotic cell cycle | $2.7 \times 10^{-10}$ |
| 13 | GO:0051054 | positive regulation of DNA metabolic process | $1.2 \times 10^{-9}$ |
| 14 | GO:0061448 | connective tissue development | $2.4 \times 10^{-9}$ |
| 15 | GO:0048146 | positive regulation of fibroblast proliferation | $2.4 \times 10^{-9}$ |
| 16 | GO:0051984 | positive regulation of chromosome segregation | $3.8 \times 10^{-9}$ |
| 17 | GO:0043491 | phosphatidylinositol 3-kinase/protein kinase B signal transduction | $2.0 \times 10^{-8}$ |
| 18 | GO:0071459 | protein localization to chromosome, centromeric region | $2.3 \times 10^{-8}$ |
| 19 | GO:0009792 | embryo development ending in birth or egg hatching | $2.9 \times 10^{-8}$ |
| 20 | GO:0016202 | regulation of striated muscle tissue development | $8.8 \times 10^{-8}$ |

[a]Enrichment analysis was performed on the top 5% genes (68 genes) in the MCC ranking shown in Table 2.

[b]The GO terms related to the biological process of *H.sapiens* were analyzed using Metascape's custom analysis.

GO: gene ontology

database (56,938 samples). To qualify, a gene had to be significantly associated ($P<0.05$) with both RFS and OS, and also show significant differential expression ($P<0.05$) between normal and tumor tissues and between tumor and metastatic tissues. As illustrated in the Venn diagram in Fig 8(A), a total of 11 genes satisfied all criteria: TOP2A, TTK, BUB1, CCNB2, KIF11, HMMR, FOXM1, FGF1, PTPN11, FGFR2, and ERBB3 (S8 Table).

We then mapped the interactions of these 11 genes with the 128 known BC biomarkers from the STRING database (Fig 8(B), S1 Table). This revealed two groups: seven genes (TOP2A, TTK, BUB1, CCNB2, KIF11, FOXM1, and HMMR) were connected to approximately 21–26% of known biomarkers, whereas the remaining four (FGF1, FGFR2, PTPN11 (SHP2), and ERBB3) were connected to only 3–5%. These connection levels may reflect the extent to which the biological functions and prognostic significance of the newly identified candidates resemble those of established biomarkers.

## Discussion

In this study, an analysis of 12 centrality measures applied to the PPI network of BC-related genes found that a high MCC score was the most consistent node feature associated with known BC biomarkers. Under the optimal network condition—using 1,500 query genes (1,362 STRING nodes) and medium confidence interactions—the proportion of known biomarkers among the top 5% of MCC-ranked genes reached approximately 36%, the highest of all evaluated conditions (Fig 4). This detection rate was approximately 1.8 times greater than that achieved with DMNC, the second-best performing metric (20.3%). The success of MCC-based biomarker detection was influenced by network parameters. First, network density was a key factor; the network constructed with medium confidence interactions was approximately 2.8- and 4.5-fold denser than those built with high and highest confidence edges, respectively, which corresponded to a 1.2- and

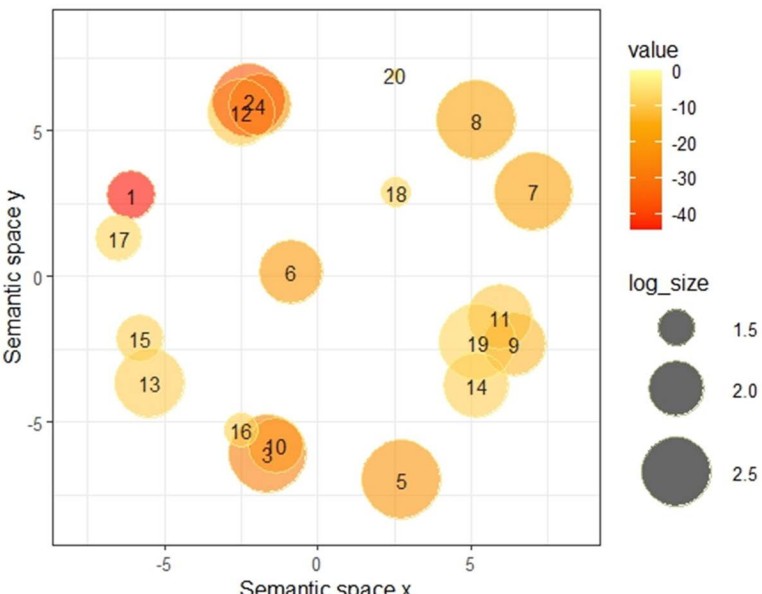

**Fig 6. REVIGO clustering of GO terms from top 5% MCC-ranked genes.** Semantically similar GO terms are positioned closely in the two-dimensional space. The number inside each bubble corresponds to the GO term list in Table 3. Bubble color indicates the log p-value, and the size indicates the term's frequency in the underlying EBI GOA database. GO: gene ontology.

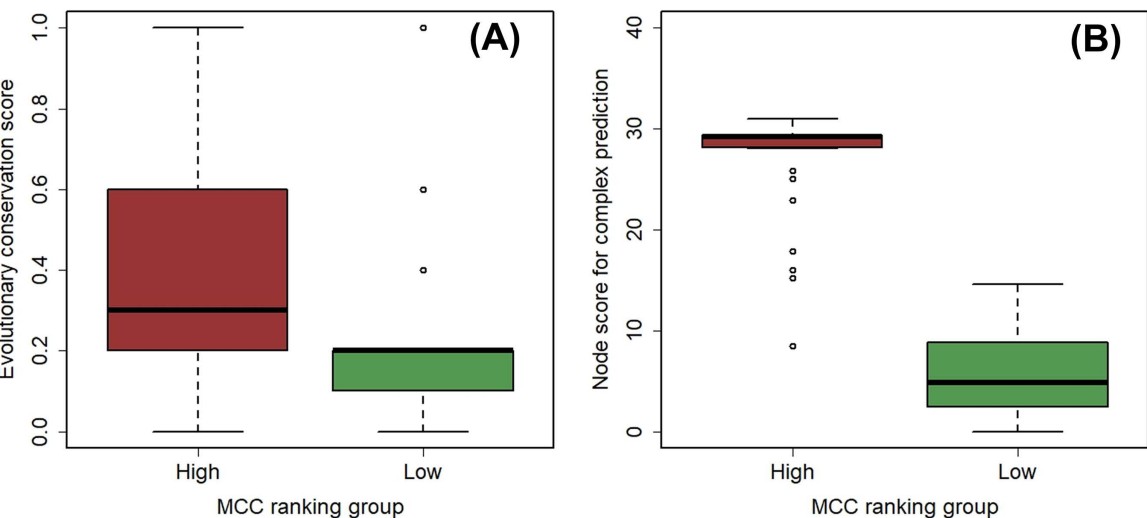

**Fig 7. Association of BC biomarker MCC ranking with biological features. (A)** Comparison of the evolutionary conservation score between high- and low-MCC ranking groups. The 64 BC biomarkers were divided into two groups, each comprising 32 genes. **(B)** Comparison of the node score for complex prediction between high- and low-MCC ranking groups. The 62 connected BC biomarkers were divided into two groups, each comprising 31 genes. HOXB13 and SLC39A were excluded from the protein complex analysis because their degree of 0 prevented the calculation of the node score.

7.7-fold higher biomarker detection rate (S3 Table, Fig 4). Another contributing factor was the number of nodes (genes), as the detection rate increased from 8.7% in the network with 1,000 query genes to approximately 36% in larger gene sets (Fig 4). These findings suggest that BC biomarkers tend to be embedded within large cliques in the PPI network,

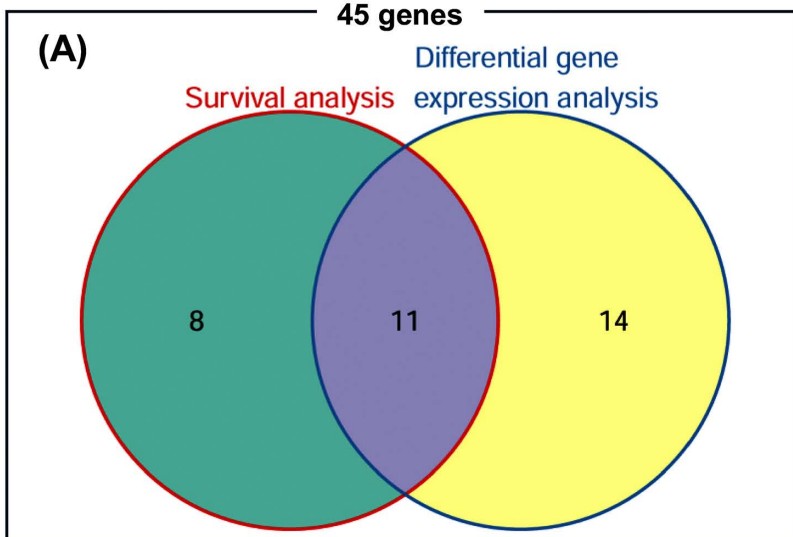

**(A)**

45 genes

Survival analysis | Differential gene expression analysis

8 | 11 | 14

**(B)**

| Key genes | MCC rank | Number of interactions with 128 biomarkers |
|---|---|---|
| TOP2A | 5 | 33 |
| TTK | 5 | 30 |
| BUB1 | 8 | 31 |
| CCNB2 | 8 | 33 |
| KIF11 | 15 | 30 |
| HMMR | 21 | 27 |
| FOXM1 | 25 | 27 |
| FGF1 | 38 | 5 |
| PTPN11 | 54 | 6 |
| FGFR2 | 63 | 4 |
| ERBB3 | 68 | 4 |

**Fig 8. Prediction of key genes as BC biomarker candidates. (A)** Venn diagram illustrating the filtering of the top 5% MCC-ranked genes based on survival and differential gene expression analyses. Among the top 68 genes in the MCC ranking (Table 2), the 45 genes that are not known biomarkers were analyzed. For survival analysis, patients were split by median gene expression, and genes with significance in both RFS and OS ($P<0.05$) were selected. For differential gene expression analysis, genes with significant expression differences ($P<0.05$) in both normal vs. tumor and tumor vs. metastatic tissues were selected. **(B)** MCC rank of the 11 identified key genes and the number of interactions with 128 known BC biomarkers. The number of direct interactions for each key gene was counted in the STRING database under medium confidence using "Experiments," "Databases," and "Co-expression" as sources. BC: Breast Cancer.

and that successful detection of these cliques requires a network that is both large and dense. In contrast, increasing the number of query genes from 1,500–2,000 resulted in a slight decrease in biomarker detection rate. One possible explanation is that the inclusion of genes with lower relevance to breast cancer (i.e., lower disease scores) led to a more diffuse interaction structure, thereby reducing the relative prominence of densely connected cliques enriched with biomarkers. As a result, the performance of MCC-based biomarker detection may have decreased.

Although BC biomarkers exhibit diverse biological functions, including hormone-related activities, cell-cycle regulation, adhesion/invasion/angiogenesis processes, epithelial and tumor-associated antigen presentation, and cellular and proliferative activity [30], we found that biomarkers and other BC-related genes with high MCC scores were predominantly associated with GO terms related to cell-cycle processes and the fibroblast growth factor receptor signaling pathway (Table 2, Fig 6). The connection to the cell cycle is particularly strong, as 19 of the 23 top-ranked biomarkers with high MCC scores have been previously reported to be involved in its regulation (S9 Table). The cell cycle is an evolutionarily conserved process, and many of its related genes are known to be highly conserved [31]. Furthermore, Wuchty et al. demonstrated that evolutionarily conserved genes are more likely to form cliques, which serve as dense interaction motifs within PPI networks [32]. Consistent with these findings, our study also demonstrated that biomarkers with high MCC scores were significantly more evolutionarily conserved (S5 Table, Fig 7(A), S7 Table). Collectively, these results suggest that evolutionarily conserved BC-related genes that are embedded in large, multiple cliques within cell-cycle-associated PPI networks may serve as sensitive biomarkers that reflect cancer progression. In this study, evolutionary conservation was assessed by assigning equal weight to each species to provide a simple and interpretable metric across diverse eukaryotes. However, we acknowledge that phylogeny-aware weighted scoring, which accounts for evolutionary distance between species, may provide a more refined evaluation of conservation. Such an approach could be particularly useful for capturing subtle differences in conservation among genes embedded in large network cliques. This represents an

important direction for future research. The tendency of cell-cycle-related genes to form cliques likely arises from their complex interaction dynamics, as many of these proteins colocalize to specific organelles, such as the centrosome, kinetochore, and midbody [33]. Approximately 20% of these proteins exhibit multi-localization, which allows them to form dynamic PPI networks centered on these proteins to regulate cell division. Additionally, many cell-cycle-related proteins are reported to function as complexes or higher-order super-complexes formed by the assembly of multiple smaller complexes, thereby exerting spatiotemporal control over their activities [33]. Our MCODE analysis supported these findings, as high-MCC biomarkers showed a strong potential for complex formation (Fig 7(B), S6 Table, S7 Table). The propensity of cell-cycle-related genes to create dynamic interaction networks and form protein complexes likely explains the formation of large and numerous cliques within the PPI network. Additionally, Metascape analysis revealed that several biomarkers (EGFR, ERBB2, FGF18, FGFR4) were also classified under the "fibroblast growth factor receptor signaling pathway" GO term, a key functional category among the top MCC-ranked genes. Receptors with this GO term are known to form heterodimers and engage with various adaptor proteins and growth factors to activate multiple signaling cascades, such as the Akt/PI3K and Ras/MAPK pathways, which would also contribute to the formation of large cliques [34,35].

Interestingly, the GO enrichment analysis of the top 5% MCC-ranked genes also identified terms such as lung development, glial cell differentiation, connective tissue development, and embryo development, which reflect fundamental biological programs related to development and differentiation (Table 3). These findings suggest that the MCC-based framework may preferentially capture genes associated with core cellular functions that are commonly dysregulated across multiple cancer types. This raises the possibility that the MCC approach could be applicable to biomarker discovery beyond BC. However, as the present study focused exclusively on a PPI network constructed from BC-associated genes, further investigation is required to determine whether this framework can be generalized to other cancer types and whether comparable biomarker detection performance can be achieved.

Experimentally validated cancer biomarkers sensitively reflect tumor initiation and progression through changes in their expression. Using a bioinformatics approach, we identified 11 genes within the top 5% of MCC-ranked nodes in the 1,500-query-gene PPI network as candidate biomarkers for BC (Fig 8(A), S10 Table). Among these, seven genes (TOP2A, TTK, BUB1, CCNB2, KIF11, FOXM1, HMMR) are connected to approximately 21–26% of the 128 known biomarkers, suggesting they may share similar prognostic characteristics (Fig 8(B)). Six of these genes (CCNB2, TOP2A, KIF11, TTK, BUB1, and FOXM1) are primarily involved in the G2/M phase of the cell cycle, promoting mitosis and regulating mitotic chromosome condensation, segregation, and bipolar spindle assembly. Moreover, their prognostic power has been reported by other researchers, supporting the validity of our bioinformatics-based candidate selection method (S10 Table). Although HMMR is also important for spindle formation, its prognostic significance remains largely unreported [36]. The remaining four genes (FGF1, FGFR2, PTPN11 [SHP2], and ERBB3) exhibited lower connectivity to the known biomarkers (approximately 3–5%), suggesting they may have distinct prognostic characteristics. Among them, FGF1, FGFR2, and PTPN11 are classified under the "fibroblast growth factor receptor signaling pathway" GO term and are involved in regulating the cell cycle and survival through the Ras/MAPK, Akt/PI3K, PLCγ, and STAT signaling pathways [37]. ERBB3 also activates the Ras/MAPK and Akt/PI3K pathways through dimerization with ERBB2, contributing to proliferation and cell survival [38]. Although ERBB3 and PTPN11 have reported prognostic potential (S10 Table), and prognostic effects of genetic variants in FGF1 and FGFR2 have been studied [39,40], to the best of our knowledge, no studies have evaluated the prognostic potential of their wild-type forms. Therefore, these genes may represent promising novel biomarkers.

Notably, the four genes that exhibited low connectivity to known biomarkers were located near the threshold of the top 5% MCC ranking (Fig 8(B)). This observation suggests that genes involved in cancer progression may participate in parallel signaling pathways that are not strongly connected to the core biomarker-enriched network and may not always rank at the very top under strict clique-based criteria. In this context, integrating MCC with centrality measures such as DMNC, which are less dependent on strict clique structures, may improve the detection of such parallel pathways that are not fully captured by MCC alone. Such integrated approaches represent an important direction for future research.

Our findings align with those of recent studies that have identified BC biomarker candidates using other bioinformatics approaches. For instance, Lin et al. identified five genes (CDC2, CCNB1, CCNA2, TOP2A, and CCNB2) as potential diagnostic biomarkers for triple-negative BC by integrating co-expression network analysis with degree centrality [41]. In this study, four of these five genes (except CDC2) were also ranked within the top 5% by MCC score (Table 2). Among them, TOP2A and CCNB2 were identified as biomarker candidates in this study, and CCNB1 is a known biomarker included in the PAM50 and OncotypeDX panels. This overlap suggests that applying the MCC score is a useful and efficient approach for selecting appropriate biomarker candidates.

However, this study had several limitations, which must be acknowledged. Notably, approximately 60% of the biomarkers did not rank within the top 5% of MCC scores in the optimal PPI network constructed from the 1,500 query genes. Therefore, future studies must identify other node characteristics shared by these biomarkers. In addition to proteins, numerous microRNAs have also been reported as potential biomarkers for the diagnosis and prognosis of BC [42]. Our next step will involve a centrality analysis on integrated PPI and mRNA regulatory networks that incorporate relationships involving microRNAs and transcription factors to evaluate the network properties of potential microRNA biomarkers in BC [43].

## Conclusion

In this study, we analyzed 12 centrality measures on a PPI network of BC-related genes, demonstrating that BC biomarkers tend to exhibit high MCC scores. In the network built with 1,500 query genes (including 64 biomarkers) and medium confidence interactions, approximately 36% of the biomarkers were found among the top 5% of genes ranked by MCC. The top MCC-ranked genes were primarily enriched in GO terms related to the cell-cycle process and fibroblast growth factor receptor signaling pathway. Furthermore, the biomarkers within this high-ranking gene set exhibited high evolutionary conservation and a strong potential for complex formation. These findings indicate that dense, cell-cycle-related regions of the PPI network contain numerous genes that can serve as reliable biomarkers capable of sensitively reflecting cancer progression. Notably, many of the 11 biomarker candidates selected using our MCC-based bioinformatic approach are related to the cell cycle and have been validated for their prognostic power in other experimental studies. The analytical approach using the MCC score, as employed in this study, is a valuable method for identifying potential cancer biomarkers.

## Supporting information

**S1 Fig. Ten types of centrality analyses.** This figure displays the results for ten types of centrality analyses (excluding degree and betweenness). The analyses were performed on a PPI network of 920 nodes, which were derived from 1,000 query genes and registered in the STRING database. Medium confidence edges (minimum required interaction score = 0.4) were used. Deeper red node colors indicate higher centrality scores, and nodes with a degree of zero are not shown. CC: Clustering Coefficient; DMNC: Density of Maximum Neighborhood Component; EPC: Edge Percolated Component; MCC: Maximal Clique Centrality; MNC: Maximum Neighborhood Component; PPI: protein–protein interaction. (TIF)

**S1 Table. BC biomarkers included in five genetic tests.** This table lists the 128 biomarkers that were selected as targets for analyzing node properties. The biomarkers are registered in the STRING database and were derived from five representative multigene tests: Mammaprint, Oncotype DX, PAM50, EndoPredict, and the BC Index. BC: Breast Cancer; PPI: protein–protein interaction. (XLSX)

**S2 Table. Algorithm for computing centrality metrics.** This table describes the five local-based and seven global-based centrality ranking methods that were applied to the constructed PPI networks of BC-related genes. BC: Breast Cancer; PPI: protein–protein interaction. (XLSX)

**S3 Table. Construction of PPI network using BC related genes obtained from the DISEASES database.** This table details the number of query genes selected from the DISEASES database based on descending disease score, along with the corresponding number of nodes registered in the STRING database. It also shows the statistics for PPI networks constructed using three different minimum required interaction scores. Network density was calculated as described in Section 2.2. (XLSX)

**S4 Table. Centrality rankings of BC biomarkers in the optimal PPI network.** This table displays the twelve centrality rankings for 64 biomarkers included in the PPI network that was constructed from 1,500 query genes (1,362 nodes, 17,907 edges, medium confidence interactions). Deeper red shading indicates a higher ranking. HOXB13 and SLC39A were excluded from the ranking because they had a degree of 0. CC: Clustering Coefficient; DMNC: Density of Maximum Neighborhood Component; EPC: Edge Percolated Component; MCC: Maximal Clique Centrality; MNC: Maximum Neighborhood Component; BC: Breast Cancer; PPI: protein–protein interaction. (XLSX)

**S5 Table. Evolutionary conservation scores for BC biomarkers.** This table presents the MCC ranking and evolutionary conservation scores (0–1.0) for the 64 biomarkers included in the 1,500-query-gene network (1,362 nodes). For each *Homo sapiens* gene, ortholog candidates across five eukaryotic species are shown. Pairs with a seed score >0.95 were considered true orthologs and are indicated in bold. An evolutionary conservation score of 0.2 was assigned for each identified true ortholog pair. BC: Breast Cancer; PPI: protein–protein interaction. (XLSX)

**S6 Table. MCODE node scores for prediction of complex formation.** This table shows the MCC rankings and node scores for 64 biomarkers from the 1,500-query-gene network (1,362 nodes, 17,907 edges, medium confidence). The node scores, which represent the capacity for protein complex formation, were calculated using the MCODE algorithm on the PPI network of BC-related genes. BC: Breast Cancer; PPI: protein–protein interaction. (XLSX)

**S7 Table. Statistical relationships between biomarker centrality and biological characteristics.** This table shows the results of the Wilcoxon rank-sum test for the evolutionary conservation and protein complex analyses. These tests evaluated the differences in evolutionary conservation scores (from S5 Table) and node scores (from S6 Table) between the high- and low-centrality ranking groups (each comprising 32 genes). CC: Clustering Coefficient; DMNC: Density of Maximum Neighborhood Component; EPC: Edge Percolated Component; MCC: Maximal Clique Centrality; MNC: Maximum Neighborhood Component; BC: Breast Cancer. (XLSX)

**S8 Table. Analysis of top 5%-MCC genes as novel biomarker candidates via survival and expression data.** This table provides the data used to identify novel biomarker candidates from the top 5% of MCC-ranked genes. Candidates were required to satisfy two conditions: (1)achieve statistical significance ($P < 0.05$) in survival analyses for both RFS and OS, and (2) show significant differential expression ($P < 0.05$) between normal and cancer tissues, and between cancer and metastatic tissues. The table lists the 11 genes that met all criteria: TOP2A, TTK, BUB1, CCNB2, KIF11, HMMR, FOXM1, FGF1, PTPN11, FGFR2, and ERBB3. RFS: relapse-free survival; OS: overall survival. (XLSX)

**S9 Table. GO categories for biomarkers with top MCC rankings.** This table lists the GO categories for the 23 known biomarkers that ranked within the top 5% for MCC. Supporting references for the functional annotations are also provided. GO: gene ontology; FGFR: fibroblast growth factor receptor. (XLSX)

**S10 Table. Functional annotation of 11 novel BC biomarker candidates.** This table presents the known biological functions and summarizes published reports on the prognostic power for each of the 11 novel biomarker candidates identified in this study. BC: Breast Cancer.
(XLSX)

## Author contributions

**Conceptualization:** Takanori Sasaki, Saito Torii.

**Investigation:** Takanori Sasaki, Saito Torii.

**Methodology:** Takanori Sasaki, Saito Torii.

**Writing – original draft:** Takanori Sasaki.

**Writing – review & editing:** Takanori Sasaki.

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
