## [Decision Letter · Decision Letter 0]

16 Mar 2026

PONE-D-26-06721Node properties of breast cancer biomarkers within a protein–protein interaction networkPLOS One

Dear Dr. Sasaki,

Thank you for submitting your manuscript to PLOS ONE. After careful consideration, we feel that it has merit but does not fully meet PLOS ONE’s publication criteria as it currently stands. Therefore, we invite you to submit a revised version of the manuscript that addresses the points raised during the review process.

We look forward to receiving your revised manuscript.

Kind regards,

Attila Csikász-Nagy

Academic Editor

PLOS One

Journal Requirements:

2. Thank you for your submission to PLOS One. We note that your cover letter mentions an article with the title "Node properties of biomarkers within the protein–protein interaction network derived from breast cancer-associated genes", but the main manuscript's title is "Node properties of breast cancer biomarkers within a protein–protein interaction". Can you please clarify which is the correct title and check if you have submitted an incorrect cover letter or incorrect manuscript file? If one is incorrect please upload the correct file.

3. Please note that PLOS One has specific guidelines on code sharing for submissions in which author-generated code underpins the findings in the manuscript. In these cases, all author-generated code must be made available without restrictions upon publication of the work. Please review our guidelines at https://journals.plos.org/plosone/s/materials-and-software-sharing#loc-sharing-code and ensure that your code is shared in a way that follows best practice and facilitates reproducibility and reuse.

4. Please note that your Data Availability Statement is currently missing the DOI/accession number of each dataset OR a direct link to access each database. If your manuscript is accepted for publication, you will be asked to provide these details on a very short timeline. We therefore suggest that you provide this information now, though we will not hold up the peer review process if you are unable.

Reviewers' comments:

Reviewer's Responses to Questions

**Comments to the Author**

1. Is the manuscript technically sound, and do the data support the conclusions?

Reviewer #1: Yes

Reviewer #2: Yes

2. Has the statistical analysis been performed appropriately and rigorously? 

Reviewer #1: Yes

Reviewer #2: Yes

3. Have the authors made all data underlying the findings in their manuscript fully available?

Reviewer #1: Yes

Reviewer #2: Yes

4. Is the manuscript presented in an intelligible fashion and written in standard English?

Reviewer #1: Yes

Reviewer #2: Yes

5. Review Comments to the Author

Reviewer #1: Thank you for this interesting and well-written study. Your approach to identifying breast cancer biomarkers through PPI network analysis is intriguing and the findings are both meaningful and well-supported. I have a few questions regarding your work and would be happy to receive your clarifications. Please find my detailed review comments uploaded as an attachment.

Reviewer #2: The paper describes a study that analyzes centrality measures on protein–protein interaction network of breast cancer related genes.

The authors explored the characteristics of known biomarkers from existing panels, as well as identified new high-ranking candidate biomarkers that may have similar prognostic value.

The paper is overall well written: the authors provide clear description of their methods, good discussion and they share the necessary information as results in tables, figures and supplemental material.

Below a few detailed comments on specific parts of the paper that need attention.

- Please provide more detailed information about the DISEASES db used, for full reproducibility purposes. DISEASES is described as a continuously updated web resource that integrates evidence on disease-gene associations from a variety of sources. Versions of the DB are archived on figshare. If the DB is accessed through a different plugin accessing the most up-to-date version of the DB, provide information on when the analysis was initiated or other information necessary to make sure that the results are reproducible.

- Authors write "we selected the top 2,000 BC-related genes with a disease score > 2.08 for our analysis". Is the 2.08 threshold selected by the authors? If so add some context to justify why this very specific number was used. If the selection was only done on the number of genes, please consider rephrasing as something like "We selected the top 2000 BC-related genes, resulting in a list with disease scores above 2.08"

- On line 112, the Kaplan-Meier citation 24 is not appropriate: please cite the relevant paper as requested by the authors of the KM online tool.

- On line 112, the Kaplan-Meier citation 25 is confusing: was the authors' intention to cite a paper about non-small-cell lung cancer?

- Figure 2, S1 and 5 are low resolution and hard/impossible to read. Please consider sharing them as SVG or other high-resolution format.

6. PLOS authors have the option to publish the peer review history of their article (what does this mean?). If published, this will include your full peer review and any attached files.

Reviewer #1: No

Reviewer #2: No

---

## [Author Response · Author response to Decision Letter 1]

27 Mar 2026

March 28, 2026

Professor Attila Csikász-Nagy

Academic Editor

PLOS ONE

Dear Professor Csikász-Nagy,

Thank you for inviting us to submit a revised draft of our manuscript entitled "Node properties of biomarkers within the protein–protein interaction network derived from breast cancer-associated genes” to PLOS ONE. We also appreciate the time and effort you and each of the reviewers have dedicated to providing insightful feedback to enhance the quality of our paper. We are glad to resubmit our article for further consideration. We have incorporated changes that reflect the detailed suggestions you provided. We hope the edits and responses we provide below satisfactorily address all the issues and concerns raised by the reviewers.

Sincerely,

Takanori Sasaki, Ph.D., Associate Professor

School of Interdisciplinary Mathematical Sciences, Meiji University, Nakano-ku, Tokyo, 164-8525, Japan

TEL: +81-3-5343-8309

E-mail: tsasaki@meiji.ac.jp

RESPONSE: We sincerely thank the editor and reviewers for their time, effort, and valuable comments on our manuscript. Your constructive feedback and suggestions have greatly improved our manuscript. Thank you for your dedication and for helping us enhance our research. The revised sentences were colored yellow.

Response to Editor Comments

Editor comment 1:

RESPONSE: Thank you for the reminder. We have reviewed the manuscript and ensured that it complies with PLOS ONE's style requirements, including file naming conventions. The formatting has been updated accordingly.

Editor comment 2:

Thank you for your submission to PLOS One. We note that your cover letter mentions an article with the title "Node properties of biomarkers within the protein–protein interaction network derived from breast cancer-associated genes", but the main manuscript's title is "Node properties of breast cancer biomarkers within a protein–protein interaction". Can you please clarify which is the correct title and check if you have submitted an incorrect cover letter or incorrect manuscript file? If one is incorrect please upload the correct file.

RESPONSE: Thank you for your comment. The correct title is "Node properties of biomarkers within the protein–protein interaction network derived from breast cancer-associated genes." We have now updated the manuscript (Lines 1-2).

Editor comment 3:

Please note that PLOS One has specific guidelines on code sharing for submissions in which author-generated code underpins the findings in the manuscript. In these cases, all author-generated code must be made available without restrictions upon publication of the work. Please review our guidelines at https://journals.plos.org/plosone/s/materials-and-software-sharing#loc-sharing-code and ensure that your code is shared in a way that follows best practice and facilitates reproducibility and reuse.

RESPONSE: Thank you for the information. This study does not rely on any author-generated code. All analyses were conducted using publicly available software tools and databases, which are described in the Methods section. Therefore, no additional code is required to be shared.

Editor comment 4:

Please note that your Data Availability Statement is currently missing the DOI/accession number of each dataset OR a direct link to access each database. If your manuscript is accepted for publication, you will be asked to provide these details on a very short timeline. We therefore suggest that you provide this information now, though we will not hold up the peer review process if you are unable.

RESPONSE: Thank you for the suggestion. We have added the information to the Data Availability Statement in the submission system and included it in the manuscript (Lines 493-499).

Editor comment 5:

RESPONSE: Thank you for the clarification. We will carefully evaluate any reviewer-recommended references and cite them where appropriate.

Editor comment 6:

RESPONSE: Thank you for the clarification. We have reviewed the reference list and made necessary corrections. All changes are described in this rebuttal letter.

Response to Reviewer #1 Comments

Reviewer #1: Thank you for this interesting and well-written study. Your approach to identifying breast cancer biomarkers through PPI network analysis is intriguing and the findings are both meaningful and well-supported. I have a few questions regarding your work and would be happy to receive your clarifications. Please find my detailed review comments uploaded as an attachment.

RESPONSE: We thank these encouraging comments.

Reviewer #1 comment 1:

In lines 217-218, the authors state that "The optimal condition was the network constructed from 1,500 query genes (1,362 STRING nodes) with medium confidence interactions, which showed the highest inclusion rate." As shown in Table 1 (line 146), the 2,000-gene network contained more biomarkers (72 vs 64) and more nodes yet performed less optimally. That is an interesting finding. Could the authors provide some insight into why a larger network did not yield better results in this case?

RESPONSE: We thank the reviewer for this insightful comment. As the number of query genes increases from 1,500 to 2,000, genes with relatively lower relevance to breast cancer (i.e., lower disease scores) are incorporated into the PPI network. These genes are likely to interact not only with clinically established biomarkers but also with other genes that may have limited relevance to prognosis. As a result, the overall interaction structure of the network may become more diffuse, reducing the relative prominence of densely connected clique structures in which biomarkers are enriched. This may lead to a relative decrease in the MCC ranking of biomarkers.

We have added a corresponding discussion in the revised Discussion section (Lines 356-360).

Reviewer #1 comment 2:

In lines 125-130, the authors state that "Pairwise comparisons were performed between H. sapiens and five other eukaryotic species" and that "an evolutionary conservation score was assigned to each biomarker, with 0.2 points awarded for the presence of an ortholog in each species, for a maximum possible score of 1.0." This is an interesting approach to measuring evolutionary conservation. Could the authors kindly share the rationale for assigning equal weight to each species? Given that D. rerio is evolutionarily much closer to H. sapiens than S. cerevisiae or A. thaliana, would a weighted scoring approach that accounts for phylogenetic distance have been considered?

RESPONSE: Thank you very much for this insightful comment. In this study, we assigned equal weight to each species when calculating the evolutionary conservation score to provide a simple and interpretable measure of cross-species conservation. Our approach was partly motivated by previous work by Wuchty et al. (2003), which demonstrated that genes participating in protein interaction motifs, such as cliques, tend to be evolutionarily conserved. In their analysis, the conservation of motif constituents did not show a clear monotonic relationship with phylogenetic distance across species, suggesting that conservation within network motifs may not strictly depend on evolutionary proximity.

Based on this perspective, we treated each species equally to capture the general tendency of conservation across diverse eukaryotes, rather than emphasizing phylogenetic distance.

That said, we agree that a weighted scoring approach incorporating evolutionary distance could provide a more refined assessment of conservation. Such an approach may better reflect subtle differences in conservation patterns, particularly for genes embedded in large network cliques. We consider this an important direction for future work and have added a corresponding note in the revised Discussion section (Lines 375–380).

Reviewer #1 comment 3:

Given that MCC inherently favors densely interconnected clique-forming genes, the authors noted in lines 343-344 that DMNC was the second-best performing metric. Since the four signaling candidates (FGF1, FGFR2, PTPN11, and ERBB3), as described in lines 331-335 and 394-398, exhibit lower connectivity to known biomarkers and appear to operate through parallel signaling pathways, could the authors kindly discuss the potential for a dual-metric ensemble approach? Specifically, would a rank aggregation of MCC and DMNC provide a more robust capture of parallel signaling pathways that might not reach the strict clique-density threshold of MCC alone, thereby strengthening the overall biomarker discovery framework proposed in this study?

RESPONSE: We thank the reviewer for this insightful and constructive comment. The four signaling-related candidate genes (FGF1, FGFR2, PTPN11, and ERBB3) exhibited relatively low connectivity to known biomarkers. As the reviewer pointed out, this observation suggests that these genes may belong to parallel signaling pathways that are not directly linked to the core biomarker-enriched network. At the same time, although these genes are included within the top 5% of MCC-ranked genes, some of them are located near the threshold. This indicates that genes involved in pathways with significant roles in cancer progression may not always rank at the very top under strict clique-based criteria. In this context, as suggested by the reviewer, integrating MCC with centrality measures such as DMNC, which are less dependent on strict clique structures, may enable more robust detection of parallel signaling pathways that are not fully captured by MCC alone. While the primary aim of this study was to evaluate the performance of individual centrality measures, we consider such integrated approaches to be an important direction for future research. This point has been added to the Discussion section (Lines 428–435).

Reviewer #1 comment 4:

In lines 432-433, the authors conclude that " The analytical approach using the MCC score, as employed in this study, is a valuable method for identifying potential cancer biomarkers." This is a significant and promising conclusion that extends beyond breast cancer. Notably, the GO enrichment analysis in Table 3 revealed "lung development" as the 9th most significant term (P = 3.7×10⁻¹³), alongside breast cancer-related processes. Could the authors kindly discuss whether this suggests the MCC-based framework is capturing a universal pan-cancer core of clique-forming genes, and whether this approach would yield comparable biomarker detection rates in other cancer types?

RESPONSE: Thank you very much for this insightful and important comment. In our GO enrichment analysis, in addition to cell cycle and fibroblast growth factor receptor signaling pathway-related terms, we also identified significant enrichment of terms such as lung development, glial cell differentiation, connective tissue development, and embryo development, which represent fundamental biological processes related to development and differentiation. These findings suggest that the MCC-based framework may preferentially capture genes associated with core cellular functions that are commonly dysregulated across multiple cancer types. Therefore, as the reviewer suggested, our MCC-based approach may have potential applicability to biomarker discovery beyond breast cancer. However, since the present study was conducted using a PPI network constructed specifically from breast cancer-associated genes, further investigation is required to determine whether this framework can be generalized to other cancer types and whether comparable biomarker detection performance can be achieved.

We have added a discussion of this point to the revised manuscript (Lines 397–406).

Reviewer #1 minor comment 1:

Could the authors please check Lines 331, and, 387 and consider changing “HMMR1” to “HMMR” to stay consistent with the official gene symbol used in the rest of the paper?"

RESPONSE: Thank you for pointing this out. We have corrected “HMMR1” to “HMMR” throughout the manuscript to ensure consistency with the official gene symbol (Lines 335, 411).

Response to Reviewer #2 Comments

Reviewer #2: The paper describes a study that analyzes centrality measures on protein–protein interaction network of breast cancer related genes. The authors explored the characteristics of known biomarkers from existing panels, as well as identified new high-ranking candidate biomarkers that may have similar prognostic value. The paper is overall well written: the authors provide clear description of their methods, good discussion and they share the necessary information as results in tables, figures and supplemental material.

RESPONSE: We thank these encouraging comments.

Reviewer #2 comment 1:

Please provide more detailed information about the DISEASES db used, for full reproducibility purposes. DISEASES is described as a continuously updated web resource that integrates evidence on disease-gene associations from a variety of sources. Versions of the DB are archived on figshare. If the DB is accessed through a different plugin accessing the most up-to-date version of the DB, provide information on when the analysis was initiated or other information necessary to make sure that the results are reproducible.

RESPONSE: Thank you for this important comment regarding reproducibility. To clarify the version of the DISEASES database used in this study, we have added the access date to the Methods section. Specifically, we now state that the database was accessed on February 23, 2024. Because DISEASES is continuously updated, this information indicates that our analyses reflect the database version available at the time of access (Lines 84-85).

Reviewer #2 comment 2:

Authors write "we selected the top 2,000 BC-related genes with a disease score > 2.08 for our analysis". Is the 2.08 threshold selected by the authors? If so add some context to justify why this very specific number was used. If the selection was only done on the number of genes, please consider rephrasing as something like "We selected the top 2000 BC-related genes, resulting in a list with disease scores above 2.08"

RESPONSE: Thank you for this helpful suggestion. We agree that the original phrasing may have implied that the disease score threshold (2.08) was explicitly selected by the authors. In fact, the gene selection was based solely on ranking, and the threshold value emerged as a result of selecting the top 2,000 BC-related genes.

We have therefore revised the sentence to: "We selected the top 2,000 BC-related genes, resulting in a list with disease scores above 2.08."to clarify this point (Lines 85-86).

Reviewer #2 comment 3:

On line 112, the Kaplan-Meier citation 24 is not appropriate: please cite the relevant paper as requested by the authors of the KM online tool.

RESPONSE: Thank you for this comment. As recommended by the developers of the Kaplan–Meier Plotter, we have replaced reference 24 with the original publication describing the tool: Győrffy B, Lanczky A, Eklund AC, Denkert C, Budczies J, Li Q, et al. An online survival analysis tool to rapidly assess the effect of 22,

---

## [Editor Report · Decision Letter 1]

5 Apr 2026

Node properties of biomarkers within the protein–protein interaction network derived from breast cancer-associated genes

PONE-D-26-06721R1

Dear Dr. Sasaki,

We’re pleased to inform you that your manuscript has been judged scientifically suitable for publication and will be formally accepted for publication once it meets all outstanding technical requirements.

Kind regards,

Attila Csikász-Nagy

Academic Editor

PLOS One
---

## [Editor Report · Acceptance letter]

PONE-D-26-06721R1

PLOS One

Dear Dr. Sasaki,

I'm pleased to inform you that your manuscript has been deemed suitable for publication in PLOS One. Congratulations! Your manuscript is now being handed over to our production team.

Kind regards,

on behalf of

Dr. Attila Csikász-Nagy

Academic Editor

PLOS One